# The Implications of Aging on Vascular Health

**DOI:** 10.3390/ijms252011188

**Published:** 2024-10-17

**Authors:** Bulbul Ahmed, Ahmed A. Rahman, Sujin Lee, Rajeev Malhotra

**Affiliations:** 1Evans Department of Medicine and Whitaker Cardiovascular Institute, Boston University School of Medicine, Boston, MA 02118, USA; ahmedb@bu.edu; 2Department of Pediatric Surgery, Massachusetts General Hospital, Harvard Medical School, Boston, MA 02114, USA; 3Division of Vascular Surgery, Department of Surgery, Massachusetts General Hospital, Harvard Medical School, Boston, MA 02114, USA; slee@mgh.harvard.edu; 4Division of Cardiology, Department of Medicine, Massachusetts General Hospital, Harvard Medical School, Boston, MA 02114, USA

**Keywords:** aging, vascular dysfunction, cellular senescence, arterial stiffness, calcification, rarefaction, and cardiovascular diseases

## Abstract

Vascular aging encompasses structural and functional changes in the vasculature, significantly contributing to cardiovascular diseases, which are the leading cause of death globally. The incidence and prevalence of these diseases increase with age, with most morbidity and mortality attributed to myocardial infarction and stroke. Diagnosing and intervening in vascular aging while understanding the mechanisms behind age-induced vascular phenotypic and pathophysiological alterations offers the potential for delaying and preventing cardiovascular mortality in an aging population. This review delves into various aspects of vascular aging by examining age-related changes in arterial health at the cellular level, including endothelial dysfunction, cellular senescence, and vascular smooth muscle cell transdifferentiation, as well as at the structural level, including arterial stiffness and changes in wall thickness and diameter. We also explore aging-related changes in perivascular adipose tissue deposition, arterial collateralization, and calcification, providing insights into the physiological and pathological implications. Overall, aging induces phenotypic changes that augment the vascular system’s susceptibility to disease, even in the absence of traditional risk factors, such as hypertension, diabetes, obesity, and smoking. Overall, age-related modifications in cellular phenotype and molecular homeostasis increase the vulnerability of the arterial vasculature to structural and functional alterations, thereby accelerating cardiovascular risk. Increasing our understanding of these modifications is crucial for success in delaying or preventing cardiovascular diseases. Non-invasive techniques, such as measuring carotid intima-media thickness, pulse wave velocity, and flow-mediated dilation, as well as detecting vascular calcifications, can be used for the early detection of vascular aging. Targeting specific pathological mechanisms, such as cellular senescence and enhancing angiogenesis, holds promise for innovative therapeutic approaches.

## 1. Introduction

Aging is associated with a gradual decline in physiological integrity, compromised functional ability, and increased susceptibility to mortality. Vascular aging is a progressive process marked by age-related deterioration in the structure and function of the heart and vasculature. Going back to the 17th century, Dr. Thomas Sydenham, a famous physician and author of Observationes Medicae, noted that “man is as old as his arteries [1]”, suggesting a longstanding acknowledgment of the vasculature’s critical role in the pathologic processes that underly aging.

Cardiovascular diseases (CVDs) are the leading cause of death for individuals over 65 [2]. As individuals grow older, their likelihood of developing serious cardiovascular conditions, such as atherosclerosis, atrial fibrillation, heart failure, and aortic stenosis, rises, due to age-related alterations in the heart and vasculature, including increased arterial thickness and stiffness, calcification, and reduced endothelial function, all of which contribute to a greater risk of cardiovascular events, such as heart attacks and strokes [3]. Over 75% of Americans aged 60–79 and nearly 86% of those over 80 suffer from CVDs [4]. Notably, the prevalence of hypertension, coronary heart disease, heart failure, and stroke increases from around 40% in the 40–59 age group to over 80% among those aged 80 years or older.

Additionally, high-socio-demographic-index countries experience delays of ten to twenty years in reaching CVD mortality rates compared to those of low socio-demographic index countries, which are more likely to have poor health literacy, reduced access to health facilities, and health care and research infrastructure that is limited in delivering adequate prevention and clinical care [5]. Additionally, the substantial decrease in CVD mortality rates observed across all age groups over the past three decades, even among countries with differing socio-demographic index levels, indicates the potential modifiability of the impact of cardiovascular aging on CVD risk [6].

Although CVD significantly contributes to age-related morbidity and mortality, many clinical trials focused on CVD to date have excluded elderly patients [7]. Animal studies have shown that therapies are less effective in older atherosclerotic animals, unlike in young and healthy animals [8,9]. In the coming decades, the increased prevalence of age-related CVDs, such as coronary artery disease, myocardial infarction, stroke, peripheral arterial disease, thoracic aortic aneurysm, valvular heart disease, and heart failure, will contribute to an even greater health and economic burden as the world’s aged population continues to grow [10].

Improving our understanding of the mechanisms that underlie age-related CVDs is essential to help facilitate early diagnosis and timely therapeutic intervention [11]. This review explores the biology and clinical manifestations of age-associated vascular dysfunction and provides a nuanced understanding of the structural and functional transformations of the aging vasculature, thereby shedding light on potential mechanisms and risk factors for age-related CVDs.

## 2. Molecular and Cellular Changes in Aging Arteries

### 2.1. Endothelial Dysfunction

The endothelium is the innermost monolayer of cells covering the luminal surface within blood vessels. It possesses dynamic properties and performs various essential functions across different segments of the arterial and organ systems [12]. Under healthy circumstances, the endothelium maintains a delicate balance between vasodilators and vasoconstrictors, oxidants, and antioxidants, as well as pro-inflammatory and anti-inflammatory, pro-thrombotic and anti-thrombotic, and pro-angiogenic and anti-angiogenic factors [12]. In the aging vascular wall, a homeostatic disruption occurs to the cellular microenvironment, resulting in endothelial dysfunction [13]. For instance, age-related reduction in endothelium-dependent vasodilation is consistently observed in coronary, brachial, and certain small arteries, which arise earlier in men than in women [14]. The phenomenon of reduced vasodilation is supported by similar findings in animal studies involving rats, rabbits, and mice [14]. Furthermore, endothelial cells from elderly mice demonstrate reduced proliferative and migratory capacities, which can result in impaired wound healing and angiogenesis [15]. These findings suggest that the mechanistic changes associated with aging occur in the absence of clinical CVD or significant risk factors, indicating that aging is an independent risk factor for endothelial dysfunction [16].

Depending on location and vessel type, age-related endothelial dysfunction manifests differently across the arterial network. The impairment of endothelial function in older individuals occurs in multiple organ systems and is associated with vasculopathies that manifest as erectile dysfunction, renal dysfunction, and retinopathy [14]. The aging-related decline in endothelial function impacts both small and large arteries, contributing to various hemodynamic alterations, including increased arterial tone in large and resistant arteries, augmented oscillatory shear stress, and enhanced large artery stiffness [17,18]. Interestingly, aging-related differences exist at the cellular level among large arteries. For instance, Luttrell et al. demonstrated that in old rats, acetylcholine-induced vasorelaxation is impaired in large conduit arteries (abdominal aorta and iliac arteries) but not in smaller conduit arteries (femoral arteries) or resistance arteries [19]. Similarly, Barton et al. did not observe aging-associated endothelial dysfunction in the femoral artery [20]. The differences in endothelial dysfunction that exist, even within the same vascular bed, underscore the complexity of mechanistic changes that occur in aging vessels.

A multitude of molecular and cellular changes occur in aged endothelial cells. In the aorta, there is an increase in end-to-end inter-endothelial junctions, while overlapping or interdigital junctions decrease with age [14]. Aging in carotid arteries results in the separation of endothelial cell–cell junctions, leading to increased permeability [21]. Studies on cerebral microcirculation reveal the loss of capillary density and the elongation of endothelial cells with age [14]. The cytoskeleton undergoes prominent reorganization, with stress fibers accumulating at the cell periphery with fewer focal adhesions [22]. Aged human umbilical vein endothelial cells exhibit an increased cell size (Figure 1) [23]. Additional important alterations in the structure and function of aged endothelial cells include deoxyribonucleic acid (DNA) damage and shorter telomere lengths [24], with some cells exhibiting polyploid nuclei, the appearance of flattened and enlarged senescent cells with reduced regenerative capacity, and the expression of several inhibitors of the cell cycle [25]. An increase in endothelial apoptosis is observed in the aorta, coronary arteries, femoral arteries, capillaries, and human umbilical vein endothelial cells with aging [14], associated with a decreased endothelial cell density [26]. Evidence indicates a decline in endothelial autophagy with age, while activating autophagy has been shown to reverse arterial aging and improve age-related endothelial dysfunction. For instance, arteries in older humans and mice exhibit impaired autophagy, reducing endothelium-dependent dilatation [27,28]. Mitochondrial dysfunction is also evident in the aging vasculature, as vascular endothelial cells exhibit impaired mitochondrial biogenesis, diminished mitochondrial mass and respiration, and elevated levels of aggregated reactive oxygen species (ROS), which directly contribute to the reduction in nitric oxide (NO) bioavailability [29].

Endothelial progenitor cells (EPCs) are a key player in maintaining endothelial function and contribute to vascular repair through their differentiation into mature endothelial cells [30]. While there are conflicting data regarding whether aging affects the total number of EPCs in humans and animals, studies consistently show that the functional capacity of circulating EPCs is impaired with advancing age [31]. Aging corresponds with diminished numbers and the self-renewal capacity of circulating EPCs across diverse subjects, accompanied by a significant decline in the colony-forming unit and migratory capacity of blood-derived EPCs [32]. Age-related changes may also impact the survival of EPCs as their telomere length decreases, potentially driving these cells into senescence [32]. Studies indicate that age-associated impairments in cardiac angiogenesis and vascular function can be partially restored by transplanting bone-marrow-derived EPCs from young individuals [33]. Moreover, the re-endothelialization capacity of transplanted old EPCs at injured sites in young animals is notably greater than that of their old counterparts, suggesting that a young niche may partially restore the diminished function of old EPCs in rats [34]. Overall, endothelial dysfunction is a hallmark of aging-related vascular diseases that is considered to be an early instigator of the complex cascade of molecular and cellular changes that manifests with vascular aging.

### 2.2. Vascular Smooth Muscle Cell (VSMC) Dysfunction

Within healthy adults, VSMCs are located in the tunica media and represent a quiescent contractile phenotype, accounting for 30% to 50% of the arterial wall volume [35]. The quantity of VSMCs in the tunica media decreases with age, and a number of VSMCs migrate from the tunica media, contributing to the thickening of the intima. The most significant change occurs during aging, marked by a phenotypic switch that involves a decrease in the expression of contractile proteins and an increase in the expression of pro-proliferative and migratory substances [36]. A similar phenotypic switch is observed during the early stages of atherosclerosis. Electron microscopy reveals the infiltration of VSMCs and the deposition of connective tissue matrix in the intima beneath an intact endothelium without any evidence of atherosclerosis in older monkeys [26]. Similar alterations in the aortic subendothelium are also evident in older humans, occurring without lipid infiltration [37]. The phenotypic switch in VSMCs observed with aging mirrors the changes seen in early atherosclerosis, suggesting a shared mechanism that contributes to vascular stiffness and plaque formation in elderly individuals.

Aged VSMCs exhibit distinct morphological irregularities and enrichment in organelles, such as the endoplasmic reticulum, Golgi apparatus, and free ribosomes [38]. These alterations reduce the active tone of VSMCs, countering the increase in wall shear stress associated with aging [39]. Evidence suggests that within aging blood vessels, VSMCs undergo senescence [40]. In humans, the VSMCs lost during aging are replaced by collagen fibers in the arterial media wall [39]. In aging rat VSMCs, an increased osteogenic tendency with an increased propensity for calcification in vitro is observed [41]. The expression of genes relevant to macroautophagy and chaperone-mediated autophagy decreases with age in aortic VSMCs [42], potentially accelerating senescence and contributing to plaque instability [43]. An upregulation of miRNA-34a is also observed in aged mouse aortas, which may increase VSMC senescence, reduce cardiac recovery from coronary artery occlusion, and increase vascular calcification [44]. Furthermore, angiotensin II signaling, including the activation of calpain-1 and matrix metalloproteinase type II, has been related to the age-associated enhancement of migration capacity in VSMCs [45]. Altogether, the age-related loss of VSMCs, enhanced osteogenic and fibromyocytic tendency, and replacement with collagen fibers compromise vascular integrity, promoting cardiovascular diseases such as atherosclerosis and hypertension.

### 2.3. Cellular Senescence

Cellular senescence is a widely recognized characteristic of aging [46]. Cellular senescence can be categorized into two types: (1) programmed senescence, which serves specific roles in early life, wound healing, and defense against tumorigenesis, and (2) unprogrammed senescence, which results from factors such as telomere length, toxic stress, and age-related damage. The accumulation of senescent cells in old age is associated with immunosenescence and exceeds the immune system’s capacity to clear these cells [47]. Identifying senescent cells is challenging because there are no specific markers that unequivocally identify all of them [48]. In order to address this challenge, researchers often utilize a combination of various markers to characterize senescent cells, including senescence-associated β-galactosidase (SAβG) activity, the expression of cyclin-dependent kinase inhibitors (p21Cip, p16Ink4a, and p53), the identification of DNA damage or critically short telomeres, and the absence of proliferation [49,50].

The senescence of vascular endothelial cells plays a crucial role in vascular aging and significantly contributes to the initiation, progression, and advancement of CVDs [51]. In typical circumstances, endothelial cells maintain a quiescent state, with only 3% of cells entering the cell cycle per month in adult mice under homeostatic conditions [52]. In aging, the proliferative capacity of endothelial cells is further restricted, where they ultimately transition into an irreversible growth arrest state referred to as senescence [53]. Senescent endothelial cells lose their ability to proliferate, become flattened and enlarged in shape and size, exhibit increased polyploidy and senescence-associated secretory phenotype (SASP) [54], and are accompanied by increased oxidative stress and a blunting of proteostasis [55]. The age-related loss of proteostasis augments ubiquitin, diminishes the activation of heat shock protein 70 (HSP70), and accumulates medin amyloid in the aorta of old mice and nearly all humans aged > 50 years [44]. Interestingly, these initial signs of endothelial senescence can also be found in some young individuals [56]. Aortic endothelial senescence results in a similar cellular enlargement and restructuring of the cytoskeleton, accompanied by a reduction in both the quantity and length of stress fibers; however, no alterations are observed in focal adhesions [22]. Both basal and shear-stress-stimulated endothelial nitric oxide synthase (eNOS) and NO production are reduced in senescent endothelial cells [57].

The expression of senescence marker p16 in human coronary arteries is positively correlated with plaque instability [58]. The senescence of endothelial cells has been observed in various other vascular organs. In the brain, endothelial senescence has been linked to increased blood–brain barrier permeability and neurovascular uncoupling [59]. The histologic analysis of post-mortem tissues revealed that atherosclerotic blood vessels contain a higher proportion of senescent endothelial and VSMCs compared to healthy arteries from individuals of the same age [60]. VSMCs in atherosclerotic plaques within the human carotid artery show increased levels of SAβG, p16, and p21, coupled with shortened telomeres [61], and a higher proportion of senescent VSMCs, recognizable by their flattened shape, atypical nuclear structure, and positive SAβG staining, were observed in older individuals [62]. Senescence in immune cells, particularly T cells, is linked to the development of atherosclerosis and serves as a biomarker for the risk of CVDs [63]. Senescent terminally differentiated CD8+ T-cells are found within unstable plaques, independently predicting all-cause mortality in the elderly [63,64]. Similar to the vasculature, various senescence cells accumulate in the heart with aging, including cardiomyocytes, endothelial cells, cardiac fibroblasts, and cardiac progenitor cells, and this accumulation is more pronounced in the myocardium of individuals with age-related CVDs [65]. In summary, a key feature of aging is cellular senescence, which involves disruptions in normal homeostatic processes, contributing to impaired immune function, vascular aging, and cardiovascular diseases.

### 2.4. Calcification

Arterial calcification is an age-related disorder characterized by the inappropriate development of osteoblastic vascular cell characteristics. Vascular calcification is defined as the deposition of calcium phosphate complexes in the vessels, which gradually transform into bone-like states. This process disrupts tissue properties, particularly elasticity, by accumulating calcium deposits in the extracellular matrix (ECM) [35]. Calcification in arteries can occur throughout a person’s lifespan, both in the intimal layer associated with atherosclerotic plaques and in the medial vessel layer, resulting in increased arterial stiffness [66]. Men typically exhibit the initial signs of calcification around age 40, while women tend to show signs around age 50 [67]. Calcification contributes to arterial stiffening and reduced distensibility, and it is linked to adverse cardiovascular outcomes [68].

The prevalence of arterial calcification increases with age. Inflammation, mitochondrial dysfunction, cellular senescence, epigenetic changes, oxidative stress, ectonucleotidases, ECM factors, and external factors (such as hyperglycemia and hyperlipidemia) also promote age-related calcification [69,70,71]. Calcifying arterial walls in the elderly have been linked to elevated cholesterol levels and genetic loci have been linked to cholesterol [71].

Arterial calcification can arise through various potential mechanisms. Bone morphogenetic protein 2 (BMP2), known for its pro-inflammatory nature, induces endothelial activation and reactive oxygen species production which further activates endothelial cells, suggesting that local inflammatory disturbances create a feedback loop, exacerbating age-related calcification [72]. BMP2 also induces the osteogenic transdifferentiation of VSMCs with the resultant deposition of calcium, implicating cellular phenotype switch in the process of calcification [73]. Increased oxidative stress is a significant factor leading to a loss of VSMC contractility, enhanced osteogenic differentiation, and calcification, characteristic of vascular aging [74]. Cellular senescence activates the SASP and innate immune signals in VSMCs, including IL-6 and BMP2, contributing to arteriosclerotic calcification [75]. The LMNA gene mutation in Hutchinson–Gilford progeria syndrome contributes to early vascular aging mechanisms, highlighting defective extracellular pyrophosphate metabolism as a factor in age-related vascular calcification [76].

Vascular calcification manifests in various forms depending on its location within the vasculature, including intimal, medial, and valvular calcification. Each type of calcification exhibits distinct yet overlapping pathophysiological mechanisms. Intimal calcification arises from factors such as lipid deposition, insulin resistance, macrophage invasion, senescence, and inflammation [77]. Medial calcification, known as arteriosclerosis, is associated with chronic kidney disease and insulin resistance, and it affects the tunica media around the internal elastic lamina, resulting in arterial stiffening and reduced blood flow due to lamellar wall hardening [78]. VSMCs in the media layer differentiate into osteoblast-like cells, secreting matrix proteins that eventually mineralize through matrix vesicle secretion or apoptosis/fibrosis [79,80]. The aging process leads to a reduction in various elastin crosslinks. As elastin undergoes degradation, elastin peptides become prone to calcification and calcium binding [81]. Valvular calcification in humans is characterized by a gradual thickening of leaflets, influenced by aging and other environmental factors. This process is marked by indicators such as inflammation, lipid deposition, neovascularization, and the emergence of ectopic mesenchymal tissue [82,83].

Calcification occurs in various arteries, including the coronary artery, carotid artery, cerebral arteries, femoral artery, iliac artery, abdominal aorta, and thoracic aorta [84]. The development of calcification in atherosclerotic lesions is associated with lipid oxidation, damage to the endothelial cells, and the subsequent inflammatory response, all factors correlated with vasculature aging [85]. Coronary artery calcium score, which is quantified from computed tomography by taking into account the area and maximal density of a coronary lesion, is well established as it correlates with coronary plaque burden and coronary heart disease events [86]. Large-scale population studies have shown an age-related increase in coronary atherosclerotic disease, evidenced by an increase in coronary calcium score [66,87,88,89], which also corresponds with an increase in the risk of sudden cardiac death [90,91]. Coronary calcification serves as a strong and independent marker for identifying vulnerable aged patients prone to fatal adverse cardiac events [87]. Studies have also demonstrated increased calcification in the carotid artery with aging [92,93]. Carotid artery calcification predicts the risk of atherosclerotic CVD, including stroke [94]. Carotid artery stiffness correlates more with thoracic aorta calcification than coronary artery calcification, likely because coronary artery calcification mainly affects the intimal layer, whereas carotid and aortic calcification involve both intima and media [95]. Although calcification across different arterial beds exhibits similar histologic characteristics and patient risk factor profiles, the genetic underpinnings vary from site to site, which speaks to the varying pathobiologies of vascular calcification [76]. Overall, arterial calcification, a hallmark of aging, is fueled by inflammation, oxidative stress, mitochondrial dysfunction, and cellular senescence. Aging-related increases in arterial calcium scores are critical indicators of coronary plaque burden and heart disease risk, highlighting the heightened susceptibility to atherosclerosis and stroke.

### 2.5. Adventitial Layer Remodeling

The adventitia regulates vascular tone and nutritional supply [14]. The main component of the adventitia is collagen and elastin fiber bundles, accompanied by a diverse range of cells [96]. These cells include fibroblasts, immunomodulatory cells, progenitor cells, as well as pericytes and adrenergic nerves [97]. The aging process in mice manifests with a higher expression of total collagen in the artery compared to young mice, primarily due to increased adventitial expression [98]. In murine carotid plaque, age-related changes in the vasa vasorum are associated with poor perfusion and increased leukocyte adhesion and extravasation, although no change was observed in angiogenic activity [99]. Scanning electron microscopy demonstrated that adventitial collagen fibers in the abdominal aorta form rope-like bundles in young aortas. However, these fibers become unraveled and flattened as they age [100]. The aging process in the aorta also facilitates the development of tertiary lymphoid organs within the aortic adventitia in elderly mice, mediated by Chemokine ligand 13 and 21 signaling pathways [101]. Furthermore, the adventitia is a site of local immune response in both aging and the early stages of atherosclerosis within the abdominal aorta, highlighting its importance in vascular aging [102].

### 2.6. Perivascular Adipose Tissue Deposition

The majority of human conduit vessels are surrounded by perivascular adipose tissue (PVAT) layers that contain a diversity of cell types, including adipocytes, pre-adipocytes, stem cells, fibroblasts, inflammatory cells (macrophages, lymphocytes, and eosinophils), nerves, and vascular cells [103]. These cells are close in proximity to the adventitial vasa vasorum and are involved in bidirectional communication with all vascular layers. Healthy PVAT maintains homeostasis by exerting anticontractile effects, offering protective benefits through the release of relaxing factors such as adiponectin, angiotensin 1–7, hydrogen sulfide, NO, and palmitic acid methyl esters [104]. Additionally, PVAT-derived adipocytokines, such as leptin, visfatin, omentin, chemerin, and resistin, play a role in regulating vascular function [104]. PVAT influences the tonus, proliferation, and migration of VSMCs, facilitating smooth muscle relaxation by transporting and storing noradrenaline [103]. PVAT induces the release of adiponectin through β3-adrenoceptor activation, resulting in PVAT-dependent vasodilation and serving as a reservoir of catecholamines released by sympathetic nerve endings to prevent smooth muscle contraction [105]. However, disruptions in PVAT function can occur in conditions such as obesity and aging, leading to pathological roles rather than protective ones [106,107]. These deleterious vascular effects include the loss of or reduction in anticontractile PVAT effects; the induction of atherogenesis, vascular hypertension, inflammation, oxidative stress, and calcification; contribution to vascular insulin resistance and inflammation; and the development of increased systolic blood pressure, pulse pressure, and age-related arterial stiffness [104,108,109]. Coronary, periaortic, and other PVATs have all been implicated in the pathophysiology of CVDs [110,111,112,113]

Increasing PVAT deposition surrounding aortic blood vessels has been observed in humans with increasing age [108]. As individuals age, coronary PVAT and adipocyte size progressively enlarge [114,115] and become more susceptible to metabolic factors that transform their function from thermogenesis to energy storage in white adipose tissue [116]. Similarly, the morphometric analysis of PVAT indicates a significant increase in the mean single adipocyte area around murine arteries during middle age [117]. Rats exhibit age-related increases in adipocyte size within abdominal PVAT and lipid deposition in thoracic PVAT [118]. Greater amounts of aortic PVAT have been associated with aging, aortic stiffness, elevated central blood pressure, increased resting aortic blood pressure, and elevated cardiovascular risk in humans [108]. Specifically, increased PVAT density is linked to a higher prevalence of atherosclerosis, culprit lesions, and cerebrovascular events in humans [108]. Preclinical animal models focused on aging and disease have provided direct evidence that PVAT surrounding various vascular beds can induce arterial stiffness, vasoconstriction, impaired arterial relaxation, and atherosclerosis [119,120,121]. Conditioned media from aortic PVAT in old mice, when compared to young mice, resulted in an increase in the intrinsic mechanical stiffness of the human aorta [108].

Several potential mechanisms may be involved in aging-associated PVAT-mediated vascular dysfunction. The aging process promotes the generation of superoxide and pro-inflammation within PVAT, contributing to increased arterial stiffness [121]. PVAT-derived superoxide is linked to arterial wall hypertrophy and the increased expression of adventitial collagen I with aging [121]. PVAT induces endothelial dysfunction through the paracrine exchange of various factors via outside-to-inside signaling in aging, including angiotensin II, chemerin, leptin, visfatin, chemokine ligand 2/3/10, nicotinamide adenine dinucleotide phosphate (NADPH) oxidase, superoxide, interferon gamma, tumor necrosis factor alpha (TNF-α), and interleukin-6 (IL-6) [103]. In obese aged mice, PVAT demonstrates heightened oxidative stress due to NADPH oxidase upregulation, reduced endothelial eNOS expression, and increased inflammation, resulting in vascular dysfunction [122]. The loss of peroxisome proliferator-activated receptor gamma coactivator 1-alpha in aged resident stromal cells within PVAT contributes to vascular remodeling by reducing brown adipogenic differentiation capacity [123]. Thoracic PVAT in aged rats exhibits increased lipid deposition and pro-inflammatory macrophages [118]. Aging-associated adipose tissue senescence, mitochondrial dysfunction, and the transition from brown to white phenotypes result in the loss of the beneficial anti-contractile properties of PVAT [124,125,126]. Furthermore, mature adipocytes and aged PVAT have been found to stimulate VSMC proliferation, exacerbating vascular dysfunction [127]. Overall, aging impairs PVAT function, which contributes to increased arterial stiffness, atherosclerosis, and heightened cardiovascular risk through pro-inflammatory and oxidative changes, amongst other disruptions in homeostasis.

## 3. Structural Changes in Aging Arteries

### 3.1. Arterial Wall Thickness

Vascular autopsies reveal that the aortic circumference increases with age, with the most significant change often observed in the ascending aorta and the least in the abdominal aorta [128]. The thickness of arterial walls significantly increases primarily due to the growth of the intimal layer, while the widths of the load-bearing medial layer and the adventitial layer remain relatively unchanged [128]. Interestingly, both the outer and inner mean axes of the tunica media gradually expand over time in the renal arterial wall [129]. These age-related changes in arterial wall thickness are not limited to central arteries but are also observed in the peripheral arteries of both the upper and lower limbs [130]. At the level of muscular arteries, as opposed to elastic arteries, the changes associated with aging are comparatively milder. However, the augmentation in diameter is more pronounced in women [131,132]. Additionally, an age-related increase in arterial wall thickness occurs in the vertebral and basilar arteries within the cerebrovascular system in humans [133].

Aging contributes to the enlargement of the carotid artery [131]. A consistently observed phenomenon is the age-related increase in intima-media thickness (IMT) in the carotid artery [134,135,136,137]. The linear increase in carotid IMT of ∼5 μm per year is consistent with advancing age in both genders [134,135]. Progressive increases in medial thickness of the carotid artery are also evident in both men and women, which distinguishes it from the aorta [138]. A murine study similarly showed that the carotid arteries of middle-aged mice exhibit smooth muscle de-differentiation and increased senescence marker expression with medial and adventitial thickening [117]. These changes in common carotid arteries result in a nearly linear increase in diastolic diameter with age, while the diastolic-to-systolic diameter ratio and peak expansion velocity decrease [139]. With these established characteristics in mind, clinical practice often relies on assessing vascular aging using carotid artery IMT, which can be non-invasively measured with ultrasonography [140].

Aging induces significant alterations in the structure and functionality of arteries, characterized by progressive thickening attributed to intimal hyperplasia due to the migration of VSMCs and the secretion of ECM molecules, including collagens and proteoglycans [128,141]. In rats, intimal collagen types I and III and elastin-like materials significantly increase with age along with the number of VSMCs [32]. Thicker arterial intimal regions with infiltrated cells and connective tissue matrix deposition are similarly evident in monkeys [26]. In humans over 65 years of age, aortic intimal thickness and cell infiltration are markedly elevated with sporadic clusters of macrophages and activated mast cells [37]. On the other hand, the increase in medial thickness primarily results from VSMC hypertrophy and an increase in collagen content [61,62]. Andreotti et al. revealed that total collagen fibers are nearly absent in the aorta at early ages while gradually accumulating with age. The study also uncovered a significant increase in total water and lipid content in the aortic wall with aging, which may explain the observed increase in wall thickness [142]. At the arteriolar level, advancing age is linked to an augmentation in the media/lumen thickness ratio, accompanied by changes in the collagen-to-elastin ratio [143]. Additionally, an age-related thickening of the capillary basement membrane has been observed in both sexes [144]. Overall, age-related intimal thickening and medial expansion are hallmarks of vascular aging and play a crucial role in the early identification of cardiovascular risk. Greater arterial wall thickness leads to increased stiffness; elevated systolic and pulse pressures; and a higher risk of left ventricular hypertrophy, heart failure, and other cardiovascular complications.

### 3.2. Arterial Diameter

An enlargement of the arterial lumen diameter is observed in aging blood vessels. Cross-sectional studies reveal that elastic proximal arteries, such as the central aorta and carotid artery, dilate with age, resulting in an augmented lumen diameter [145,146]. Corresponding to the changes in carotid lumen diameter with age, muscular arteries also exhibit an increase in lumen diameter, with a more pronounced effect observed in women than men [138,147]. On average, the luminal diameter of the thoracic aorta increases by 1.5 to 1.7 mm every ten years [148]. The correlation between age and increasing arterial lumen diameter extends to peripheral arteries, including brachial, popliteal, femoral, and radial arteries [149]. The increasing luminal diameter of the common carotid arteries is independently associated with elevated IMT and atherosclerotic plaques [138]. The enlargement of arterial lumen diameter in aging is attributed to elastic degradation, lipid infiltration, and the repetitive stretching of elastic arteries throughout a person’s lifetime, leading to elastin fatigue, fracture, and fragmentation [150,151]. In small vessels, including the arterioles and capillaries, increasing age is associated with decreased lumen diameter [152].

The Framingham Heart Study has established a reference value for aortic root diameter based on echocardiography in a large healthy population. The study found that the aortic root diameter consistently grows with age, from 28–33 mm at 25 years old to 33–37 mm at 75 years old [153]. Other echocardiography-based studies have shown that the ascending aorta and aortic root diameter are closely related to age. Even after body surface area adjustment, the aortic root diameter steadily increases [154,155]. A Caucasian population study found a 1.1 mm increase in the diameter of the aortic root and a 0.9 mm increase in the ascending aorta diameter per decade in individuals 15 years and older [156]. Furthermore, significant differences in carotid diameter between males and females have been observed from 25 years of age and onwards. Carotid diameter and pulsatile diameter exhibit age- and sex-related changes. Males demonstrate a marked increase in carotid diameter with age, while females exhibit a more modest increase. Specifically, the carotid diameter in males increased by 2.1 mm from 15 to 70 years of age, compared to a 0.8 mm increase in females [157]. When assessing the wall diameter of atherosclerosis-prone (e.g., carotid artery, femoral, superficial femoral, and popliteal artery) and atherosclerosis-resistant (e.g., brachial artery) conduit arteries, larger diameters have been associated with older age across all arteries [137]. Notably, muscular arteries, in contrast to elastic arteries, show milder alterations with age. The diameter of muscular arteries increases with age, particularly in women [132]. Overall, aging results in an increased arterial diameter and reduced elastic recoil, leading to aortic dilation and elevated stiffness. This increased load on the aging heart elevates wall tension, alters hemodynamics, reduces cardiac efficiency, and contributes to left ventricular hypertrophy and hypertensive heart disease [158].

### 3.3. Arterial Stiffness

Multiple epidemiological investigations have demonstrated that the rigidity of arteries predicts the onset of CVD in the general population [159]. Aortic stiffness exhibits a dramatic increase with age in the majority of individuals. Carotid–femoral pulse wave velocity is markedly increased with age and strongly correlates with major cardiovascular events, especially in younger adults [160]. In the Framingham Heart Study, the prevalence of high-risk aortic stiffness is less than 1% before age 50 but exceeds 60% after reaching 70, particularly among women [161]. Stiffness in large elastic arteries is predominantly influenced by components within the ECM, such as the elastin-to-collagen ratio, which decreases towards the periphery, thereby resulting in higher arterial stiffness in more distal arteries. The rigidity of smaller arteries and arterioles is controlled by vascular wall thickening and smooth muscle tone [162].

Aging leads to changes in the elements of the aortic wall, with a particular impact on elastin and collagens. These two substances make up the majority of the ECM present in arteries. With age, there is a reduction in VSMCs in the medial layer and increased fragmentation of elastic fibers, accompanied by an increase in collagen fibers. Consequently, the mechanical load on the artery shifts towards collagen fibers, which are 100–1000 times stiffer than elastic fibers [163,164]. Elevated collagen in arteries leads to increased rigidity, both structurally and functionally, as individuals age. Collagen types I and III are predominantly found in the aorta, and their concentration gradually rises after age 50 [165]. Consequently, the collagen content undergoes crosslinking through enzymatic (via lysyl oxidases) and non-enzymatic (via advanced glycation end products) mechanisms, further elevating vascular stiffness [166]. Collagens surrounding VSMCs progressively undergo glycation, reinforcing the stiffening of VSMCs. Aged VSMCs therefore exhibit increased adhesion strength [38].

Glycoproteins and integrins play a crucial role in facilitating the anchoring of VSMCs to elastic lamellae [167] that enable the vessel to endure the blood pressure generated by the heart and protect the vascular structure from excessive load. However, in aging arteries, elastin distribution becomes heterogeneous, lacking coherent orientation in the inner media [168]. A hallmark feature of arterial aging is the gradual thinning, splitting, fraying, and fragmentation of elastic lamellae [169]. As individuals age, the intima undergoes thickening due to the breakdown of internal elastic fibers and the deterioration of the elastic lamellae in the media, while the elastic fibers become irregular and fragmented [170]. Age-related arterial structural remodeling is significantly influenced by the chronic activation of metalloproteases, leading to elastin fiber degradation and promoting VSMC migration by detaching cells from the ECM [171]. The structure of elastic lamellae also undergoes damage with age due to repeated mechanical loading, continuous cycles of stretching and recoiling, and oxidative stress, resulting in fragmentation, rupture, and diminished vascular distensibility. Simultaneously, as elastin can undergo non-enzymatic glycation and calcification, collagen molecules progressively acquire crosslinks, contributing to increased arterial stiffness [172].

Several other factors contribute to increasing vascular stiffness, including calcium deposition, impaired endothelial function, and alterations in the structure and function of VSMCs. The aging process exacerbates endothelial impairment and reduces NO bioavailability, culminating in a pro-inflammatory and vasoconstrictive state, thus elevating vascular fibrosis and arterial stiffness [173]. Additionally, endothelial dysfunction in aging induces oxidative stress through increased superoxide production, causing vascular structural damage and hemodynamic alteration, leading to stiffness [12]. Furthermore, aging endothelium experiences diminished autophagy, a cellular housekeeping mechanism crucial for maintaining homeostasis, resulting in increased oxidative stress, which may induce arterial stiffness [27].

Arterial stiffening, particularly in large arteries, is proposed to impact CVDs in various ways. Firstly, elastic aortas function as capacitance vessels, mitigating pressure spikes during systole by expanding and recoiling during diastole to ensure continuous blood flow, known as the Windkessel effect [174]. However, with aging, the aorta loses its elasticity and becomes stiffer, impairing the capacitance function and leading to increased systolic blood pressure. Secondly, increased arterial stiffness accelerates the velocity of the pressure waveform, impacting the return of reflected waves and amplifying systolic pressure, adding workload to the left ventricle which can result in adverse ventricular remodeling. Lastly, increased stiffness leads to elevated systolic pressure, reduced diastolic pressure, and widened pulse pressure, potentially contributing to organ damage, particularly in vulnerable regions such as the cerebral and renal circulations [12]. In summary, arterial stiffness, which increases with age, is a major predictor of CVD risk. Elevated collagen content and reduced elastin in the arterial wall contribute to this stiffness, impairing the vessel’s ability to function as an effective capacitance vessel. This leads to exacerbated systolic blood pressure and the potential for adverse cardiac remodeling.

### 3.4. Rarefaction of Collaterals

Collateralization, the emergence of a vessel from a pre-existing one, is crucial in determining the severity of tissue damage following acute arterial obstruction. The aging process induces collateral rarefaction, diminishes the conductance of the collateral network, and impairs collateral remodeling and perfusion in ischemic conditions that increase the resistance of the collateral circulation and exacerbate ischemic stroke outcomes [13,175]. In particular, the absence of collaterals is an independent predictor of mortality in elderly patients over 70 years [176]. Collateral blood vessels typically take around 20 days to form following an ischemic event [177]. However, age-related impairments in the capacity of collateralization and perfusion recovery are observed in coronary arteries following myocardial infarction [177]. Only one-third of elderly patients develop sufficient angiogenic growth to adequately supply blood to the myocardium after occlusion [178]. A human study showed that aging leads to a decline in the number and diameter of collaterals in skeletal muscle, resulting in substantial decreases in blood flow and increased tissue injury post-occlusion, as well as inadequate collaterals in the brain with a six-fold increase in collateral resistance and a three-fold increase in the severity of infarct volume following middle cerebral artery occlusion in old mice [13]. Furthermore, collateral blood vessels formed in aging individuals display increased tortuosity compared to their younger counterparts [13].

Capillary rarefaction is a reduced density of microvascular networks, including arterioles and capillaries [13]. Aging is associated with arteriolar rarefaction, a decrease in capillary density, and vessel loss. This can, for instance, lead to diminished cerebral blood flow and changes in the structure of the remaining vessels, ultimately reducing metabolic support for neuronal signaling, particularly during increased neuronal activity in both animals and humans [179]. In the retina, vascular rarefaction exhibits an inverse correlation with age [180]. In the skin, capillary density decreases with aging and other conditions, such as chronic heart failure, hypertension, and renal failure [181]. Aging effects lead to the rarefaction of coronary arterioles, impairing coronary microvascular function in coronary artery disease subjects [176,182]. Capillary rarefaction also occurs due to premature vascular aging, leading to arterial stiffening, endothelial dysfunction, and increased central pulse pressure in children and young adults, resulting in target organ damage and cardiovascular events [183].

The impairment of angiogenesis and vasculogenesis associated with aging plays a central role in collateral formation and microvascular rarefaction [175]. Several factors contribute to this age-related phenomenon, including the angiogenic incompetence of endothelial cells, the reduction in angiogenic stimuli, pericyte and VSMC dysfunction, and the upregulation of angiogenesis inhibitors [181]. The initiation of angiogenesis and the formation of early vascular structures depend on endothelial cells [184]. As endothelial cells age, their capacity for angiogenesis decreases, which impairs microvascular homeostasis, ischemic tissue revascularization, and the healing of tissue injuries [185]. Primary microvascular endothelial cells derived from aged rodents exhibit impaired responses to exogenous vascular endothelial growth factor (VEGF) administration [175]. In the vasculature, apoptotic endothelial cells exhibit reduced NO bioavailability, increased TNF-α expression, and increased mitochondrial oxidative stress, collectively preventing angiogenesis, contributing to age-related microvascular rarefaction across organ systems, including the heart, kidney, and skin [31]. Furthermore, the increased presence of senescent endothelial cells or VSMCs in aging vasculature contributes to impaired angiogenesis and microvascular rarefaction [186]. Experimentally induced cellular senescence in the brain promotes cerebromicrovascular dysfunction, leading to microvascular rarefaction and cognitive deficits, resembling the aging phenotype of the brain [187,188]. The decline in the production of EPCs may also play a pivotal role in vascular rarefaction [189]. Pericytes also play a critical role, as bidirectional signaling between pericytes and endothelial cells is necessary for maintaining the integrity of capillary flow and structure. The loss of pericytes during aging contributes to the phenomenon of rarefaction in the aged brain [190].

As individuals age, there is a diminished capacity to repair and form blood vessels, partially attributed to a reduced ability to generate VEGF [191]. Studies have shown that introducing exogenous VEGF or restoring Hif1α expression can mitigate the decline in limb blood flow recovery in aged mice, though clinical benefit has not been proven in humans with peripheral artery disease [192,193,194]. Platelet-derived growth factor (PDGF) facilitates blood vessel maturation by recruiting smooth muscle cells and pericytes [195]. However, PDGF levels in plasma decrease after the age of 25 [196], which may contribute to vascular rarefaction. Additionally, the reduced bioavailability of endothelium-derived NO disrupts the dynamic balance between angiogenesis and vascular regression, promoting microvascular rarefaction and insufficient collateral circulation in various tissues, exacerbating ischemic tissue injury with aging [13,197]. Genetic eNOS deficiency also accelerates collateral vessel rarefaction and impairs the activation of arteriogenesis in young mice [198]. Overall, elderly individuals exhibit diminished collateral vessel formation, which exacerbates the severity of ischemic events, such as acute arterial obstruction, by impairing vascular repair and increasing tissue damage post-ischemia. This diminished capacity contributes to a heightened risk of cardiovascular events and poorer outcomes in myocardial infarction and stroke, underscoring the need for targeted therapeutic strategies to enhance angiogenesis and vascular repair in older adults.

## 4. Therapeutics and Future Research

Currently, there are no clinical therapies specifically designed to prevent or reverse vascular aging. However, a healthy lifestyle, including a balanced diet, regular exercise, caloric restriction, weight loss, and managing risk factors (such as hyperlipidemia, diabetes, and hypertension), has been shown to partially mitigate vascular aging and related cardiovascular events [199,200].

Sedentary lifestyle is positively associated with vascular aging, while physical exercise improves endothelial function and is associated with reduced arterial stiffness as people age [201,202,203]. A 10-year longitudinal study found that regular exercise delayed the age-related increase in brachial–ankle pulse wave velocity [204]. Exercise also benefits mitochondrial function and antioxidant production in older animals [205], and, in older men, it has been shown to enhance the number and function of endothelial progenitor cells [206].

Caloric restriction has been shown to significantly slow down vascular aging, extending lifespan and protecting against cardiovascular disease in rodents and nonhuman primates [207,208,209,210]. In a murine model, caloric restriction improved nitric oxide bioavailability, reduced oxidative stress and inflammation, and promoted anti-aging microRNA expression in endothelial cells [208,211,212]. However, in patients with multiple cardiovascular risk factors, the benefits of caloric restriction are less evident [213].

Senolytic therapies, including quercetin, dasatinib, navitoclax, nutlins, and UBX0101, have demonstrated potential in treating age-related diseases and extending lifespan in aged mice [214,215]. A combination of quercetin and dasatinib effectively clears senescent cells, reduces the number of naturally occurring senescent cells and their release of frailty-associated proinflammatory cytokines in human adipose tissue explants, and extends healthy lifespan in mouse models [216]. Chronic treatment with these compounds reduced senescent cells in the aorta, improved nitric oxide bioavailability, alleviated vasomotor dysfunction, and reduced aortic calcification in aged atherosclerotic mice. A phase 1 clinical trial further supports these findings, showing that a 3-day oral treatment with quercetin and dasatinib significantly reduced the burden of senescent cells in older diabetic kidney patients [217]. Similarly, fisetin has demonstrated senotherapeutic effects and has reduced senescence markers in multiple tissues in mice and in human adipose tissue [218].

Age-related increases in circulating carbohydrates and lipids with associated age-related insulin resistance are major contributors to cardiovascular complications. Addressing these risk factors through a balanced diet and regular exercise can greatly reduce the risk of cardiovascular diseases and improve overall metabolic health [219]. It remains to be determined what the specific effects of newer agents in the treatment of diabetes and obesity, such as GLP-1 agonists and DPP-4 inhibitors, are on vascular aging [220,221].

Future research efforts will need to focus on understanding the interplay between aging and chronic diseases, with an emphasis on aligning preclinical vascular aging studies with human clinical investigations. In addition to targeting senescent cells, vasculo-metabolic pathways, caloric restriction, and physical exercise, therapies aimed at promoting angiogenesis could be considered, as aging-related angiogenic dysfunction and microvascular rarefaction are seen in aged coronary artery diseases subjects [222]. Potential molecular targets, such as mTOR, sirtuins, and AMPK, which are dysregulated in aging, also offer promising avenues for pharmacological interventions [223,224,225]. Overall, the complex processes driving vascular aging will likely necessitate a more nuanced, individualized, and multi-targeted approach to therapy to derive the greatest clinical benefits.

## 5. Conclusions

This review offers insights into the complex interplay of the vascular system’s molecular, cellular, structural, and physiological processes associated with aging. As individuals age, the body progressively develops vascular pathologies, such as arterial fibrosis, stiffness, rarefaction, distensibility, fat deposition, and calcification, thereby increasing susceptibility to serious cardiovascular events. Understanding the complex processes of vascular changes in aging may serve as a foundation for further research and interventions to improve cardiovascular function. This knowledge may pave the way for new therapies that target the fundamental aging process, helping to ameliorate vascular diseases.

## Figures and Tables

**Figure 1 ijms-25-11188-f001:**
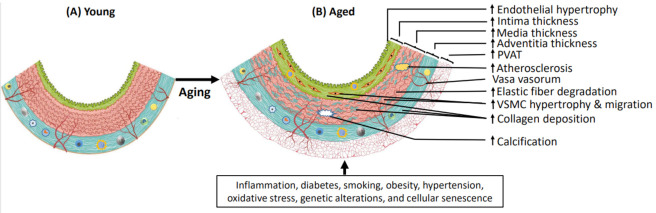
The aging-associated structural alterations in large arteries. Unlike younger arteries, the aged vasculature undergoes significant changes characterized by elastin degradation and collagen accumulation, leading to increased stiffness, thickness, and decreased elasticity. Age-related intimal thickening is consistently observed across all large arteries, while the carotid artery is notable for thickening in all the intima, media, and adventitia layers. There is a noticeable expansion in the lumen diameter and evident fat deposition on the adventitia. Calcification also occurs in both the intima and medial layers of large arteries. At the cellular level, changes associated with the hypertrophy, senescence, and apoptosis of VSMCs and endothelial cells are evident. VSMCs migrate from the media to the intima and exhibit dysfunctional characteristics. Endothelial dysfunction is also apparent, which significantly reduces vasodilation and neo-vascularization in the elderly. Factors such as inflammation, diabetes, smoking, obesity, hypertension, oxidative stress, and unhealthy lifestyle choices can further exacerbate vascular aging.

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
