# Peer review of "The Implications of Aging on Vascular Health"

_ijms, 2024, doi:10.3390/ijms252011188_

Round 1

Reviewer 1 Report

Comments and Suggestions for Authors

I have received for review an article entitled ,,The implications of aging on vascular health” which is being processed by International Journal of Molecular Sciences.

The proposed manuscript is one with therapeutic and prognostic impact, but I encourage authors to address the following issues to improve its quality:

Abstract -summarizes the main elements analyzed in the manuscript, emphasizing the prognostic implications in terms of associated cardiovascular risk. I encourage authors to specify more keywords in order to increase the dissemination rate of the manuscript

Introduction - mentions the main existing data in the specialized literature related to the topic of the manuscript. Epidemiologic data were provided, but it would be useful to mention the risk associated with aging in the main CV pathologies.

The presented sections explain the main pathophysiologic mechanisms associated with aging and increased cardiovascular risk, but I suggest the introduction of additional sections:

- Aging and management of lipid and carbohydrate profile - focusing on intricate pathophysiological mechanisms

- future research directions

- impact of therapies associated with slowing biological aging on vascular and cardiovascular prognosis

I congratulate the authors for the iconography. However, a central image summarizing the main pathophysiological mechanisms associated with vascular aging would be useful.

Rows 616-623 - missing information as per journal recommendations

In conclusion, the proposed manuscript brings to attention an extremely interesting topic, presenting scientific information with therapeutic and prognostic value, but needs revision in order to be considered for publication.

Author Response

Comments 1: The proposed manuscript is one with therapeutic and prognostic impact, but I encourage authors to address the following issues to improve its quality:

Abstract -summarizes the main elements analyzed in the manuscript, emphasizing the prognostic implications in terms of associated cardiovascular risk. I encourage authors to specify more keywords in order to increase the dissemination rate of the manuscript

Response 1: Abstract: We appreciate your suggestion to emphasize the prognostic implications in terms of cardiovascular risk. In response, we have revised the abstract to better highlight these aspects and ensure the focus on associated cardiovascular risk is more prominent.

Keywords: In line with your recommendation, we have added more specific keywords to increase the visibility and dissemination of the manuscript. The additional keywords include: cellular senescence, arterial stiffness and thickness, calcification, rarefaction. These new terms should help improve discoverability across a wider audience.

Comments 2: Introduction - mentions the main existing data in the specialized literature related to the topic of the manuscript. Epidemiologic data were provided, but it would be useful to mention the risk associated with aging in the main CV pathologies.

Response 2: Thank you for your insightful comment on the introduction.

In response, we have revised the introduction to include a discussion on the risk associated with aging in cardiovascular (CV) pathologies. We have added, “As individuals grow older, their likelihood of developing serious cardiovascular conditions such as atherosclerosis, atrial fibrillation, heart failure, and aortic stenosis rises, due to age-related alteration in the heart and vasculature, including increased arterial thickness and stiffness, calcification, and reduced endothelial function, all of which contribute to a greater risk of cardiovascular events like heart attacks and strokes [3]”.

Comments 3: The presented sections explain the main pathophysiologic mechanisms associated with aging and increased cardiovascular risk, but I suggest the introduction of additional sections:

- Aging and management of lipid and carbohydrate profile - focusing on intricate pathophysiological mechanisms

- future research directions

- impact of therapies associated with slowing biological aging on vascular and cardiovascular prognosis

Response 3: Thank you for your valuable suggestions regarding the manuscript.

In response to your feedback, we have added a separate section titled "Therapeutics and Future Research," which addresses future research directions and discusses the impact of therapies associated with slowing biological aging on vascular and cardiovascular prognosis. While we did not introduce a distinct section on aging and the management of lipid and carbohydrate profiles, we have incorporated relevant discussions throughout the manuscript to enhance clarity on these intricate pathophysiological mechanisms.

Comments 4: I congratulate the authors for the iconography. However, a central image summarizing the main pathophysiological mechanisms associated with vascular aging would be useful.

Response 4: Thank you for your kind words regarding the figure in our manuscript. We appreciate your suggestion to include a central image summarizing the main pathophysiological mechanisms associated with vascular aging. In response, we have revised the existing figure to incorporate these mechanisms. The updated figure now provides a more comprehensive summary of the key processes, ensuring clarity while avoiding redundancy.

We believe the revised figure effectively captures the essential mechanisms discussed, and we hope it meets your expectations. We appreciate your valuable input and remain open to any further suggestions.

Comments 5: Rows 616-623 - missing information as per journal recommendations

Response 5: Thank you for pointing this out. Upon reviewing lines 616-623, we confirm that these lines contain references formatted according to the journal’s guidelines. All references were managed using EndNote software, ensuring they are correctly cited and in alignment with the journal’s requirements.

If there are any specific issues or discrepancies that we may have overlooked, please feel free to provide further clarification, and we will address them promptly.

We appreciate your attention to detail and look forward to your feedback.

Comments 6: In conclusion, the proposed manuscript brings to attention an extremely interesting topic, presenting scientific information with therapeutic and prognostic value, but needs revision in order to be considered for publication.

Response 6: Thank you for your thoughtful and constructive feedback on our manuscript. We are pleased to hear that you found the topic scientifically valuable with therapeutic and prognostic implications.

In response to your recommendation, we have conducted revisions throughout the manuscript to improve clarity, ensure consistency, and further strengthen the content. These revisions include refining specific sections, enhancing the discussion of key points, and improving the overall presentation to meet the journal’s standards.

We believe that these revisions have addressed the necessary improvements, and we are hopeful that the manuscript is now suitable for publication.

Reviewer 2 Report

Comments and Suggestions for Authors

Bulbul et al provides a detailed overview of the existing literature on the impact of aging on vascular health in the current manuscript. It covers a wide range of molecular and cellular alterations as well as structural changes that occur in the aging vasculature, which are all critical in understanding vascular aging. The manuscript is generally well-written and informative, making a valuable contribution to the field of vascular biology. Some suggestons:

1. some cited works date back more than a decade. It is crucial to ensure that all references are current and reflect the most recent findings within the discipline.

2. there are minor inconsistencies in the structure of the manuscript, with certain parts appearing to diverge from the main narrative.

3. A typographical error?: in line 40, the phrase "[m]an is as old as his arteries" appears with a brackets around "m". Is it a typographical error, and should it be "man is as old as his arteries" without the brackets?

Author Response

Comments 1: Bulbul et al provides a detailed overview of the existing literature on the impact of aging on vascular health in the current manuscript. It covers a wide range of molecular and cellular alterations as well as structural changes that occur in the aging vasculature, which are all critical in understanding vascular aging. The manuscript is generally well-written and informative, making a valuable contribution to the field of vascular biology. Some suggestons:

  1. some cited works date back more than a decade. It is crucial to ensure that all references are current and reflect the most recent findings within the discipline.

Response 1: Thank you for your positive feedback on our manuscript and for highlighting its contribution to the field of vascular biology.

In response to your suggestion regarding outdated citations, we have thoroughly reviewed the references throughout the manuscript. As a result, we have deleted older citations and replaced them with more recent, relevant studies that reflect the latest advancements in the discipline. The updated citations have been incorporated into the respective sections to ensure the manuscript is both current and aligned with the most recent findings in vascular aging.

We appreciate your careful review and believe that these changes strengthen the manuscript.

Comments 2: there are minor inconsistencies in the structure of the manuscript, with certain parts appearing to diverge from the main narrative.

Response 2: Thank you for your feedback and for bringing this to our attention.

In response to your observation regarding minor structural inconsistencies, we have carefully revised the manuscript to ensure that all sections are aligned with the main narrative. We have made adjustments to sections that appeared to diverge, refining the flow and coherence throughout the manuscript to strengthen the overall structure.

We believe these revisions have improved the readability and clarity, allowing the key points to be more clearly presented in the context of the broader narrative.

Comments 3: A typographical error?: in line 40, the phrase "[m]an is as old as his arteries" appears with a brackets around "m". Is it a typographical error, and should it be "man is as old as his arteries" without the brackets?

Response 3: Thank you for pointing out the typographical error in line 40. We appreciate your attention to detail. We have corrected the phrase to read "Man is as old as his arteries" as you suggested.

Round 2

Reviewer 1 Report

Comments and Suggestions for Authors

The proposed manuscript has been improved and can be considered for publication. I congratulate the authors for their effort in analyzing such a large number of references and producing an update in the vascular field. This version of the manuscript no longer meets the formatting conditions imposed by the journal. Please review the formatting.